# Choline and Betaine Levels in Plasma Mirror Choline Intake in Very Preterm Infants

**DOI:** 10.3390/nu15224758

**Published:** 2023-11-12

**Authors:** Michaela Minarski, Christoph Maas, Christine Heinrich, Katrin A. Böckmann, Wolfgang Bernhard, Anna Shunova, Christian F. Poets, Axel R. Franz

**Affiliations:** 1Department of Neonatology, University Children’s Hospital, Tübingen University Hospital, 72076 Tübingen, Germanywolfgang.bernhard@med.uni-tuebingen.de (W.B.);; 2Center for Pediatric Clinical Studies, University Children’s Hospital, Tübingen University Hospital, 72076 Tübingen, Germany

**Keywords:** infant, preterm, very low birth weight infant, enteral feeding, nutrition, breast milk, choline, betaine

## Abstract

Choline is essential for cell membrane formation and methyl transfer reactions, impacting parenchymal and neurological development. It is therefore enriched via placental transfer, and fetal plasma concentrations are high. In spite of the greater needs of very low birth weight infants (VLBWI), choline content of breast milk after preterm delivery is lower (median (p25–75): 158 mg/L (61–360 mg/L) compared to term delivery (258 mg/L (142–343 mg/L)). Even preterm formula or fortified breast milk currently provide insufficient choline to achieve physiological plasma concentrations. This secondary analysis of a randomized controlled trial comparing growth of VLBWI with different levels of enteral protein supply aimed to investigate whether increased enteral choline intake results in increased plasma choline, betaine and phosphatidylcholine concentrations. We measured total choline content of breast milk from 33 mothers of 34 VLBWI. Enteral choline intake from administered breast milk, formula and fortifier was related to the respective plasma choline, betaine and phosphatidylcholine concentrations. Plasma choline and betaine levels in VLBWI correlated directly with enteral choline intake, but administered choline was insufficient to achieve physiological (fetus-like) concentrations. Hence, optimizing maternal choline status, and the choline content of milk and fortifiers, is suggested to increase plasma concentrations of choline, ameliorate the choline deficit and improve growth and long-term development of VLBWI.

## 1. Introduction

Choline is essential for the synthesis of the predominant phospholipids phosphatidylcholine and sphingomyelin of cell membranes, for the formation of secretions like bile, surfactant and lipoproteins and for the synthesis of the neurotransmitter acetylcholine [1,2]. Choline and its metabolites like acetylcholine play a crucial role in fetal and neonatal neurogenesis, brain organization and the formation of synapses [2,3,4,5,6]. Moreover, the choline metabolite betaine, accounting for ~40% of choline metabolites, is important for osmotic regulation and kidney function and provides methyl groups to form methionine from homocysteine. In its active form, i.e., S-adenosyl methionine (SAM), it represents the most important methyl group donor for metabolism [7,8,9]. Since choline is the major source of betaine, their plasma concentrations correlate with each other [10]. Because of the central role of choline in many metabolic pathways, liver function and neurologic development, choline was recognized as an essential nutrient in 1998 by the Institute of Medicine (IoM) of the USA, and in 2016 by the European Food Safety Authority (EFSA) [11]. This holds in spite of endogenous choline synthesis through the PEMT pathway, as it is insufficient to fulfil total choline requirements [12] and sufficient supply during rapid growth seems indispensable [13].

Free, water-soluble choline, due to its high metabolic rate and rapid intracellular uptake, accounts for barely 1% of the total plasma choline pool. However, its availability is crucial to enable the described metabolic processes and the formation of phosphatidylcholine (main fraction of the plasma pool with approx. 88%).

The fetus shows a particularly high growth rate between the 24th and 34th week of pregnancy [13,14,15] and, accordingly, has a high nutrient requirement during this period. Notably, total choline content of an organism is proportional to body mass, and requirements are proportional to its growth rate [10,13]. In order to meet these needs in terms of choline supply, the latter is actively transported via the placenta to the fetus [16,17], so that fetal choline plasma levels are about three times higher than maternal levels, even though the latter also slightly increase during pregnancy [18]. These elevated maternal choline concentrations are related to upregulation of the PEMT pathway by high estrogen levels during pregnancy [19]. Despite the higher adequate intake recommendations for pregnant women, they mostly become choline-depleted by end-gestation [20]. Moreover, in 25–45% of the West-European population, single nucleotide polymorphisms (SNPs) of the PEMT gene blunt endogenous choline formation, accentuating increased exogenous requirement [21,22,23].

In case of premature birth, the trans-placental supply of choline via the umbilical cord is suddenly interrupted, so that plasma levels of preterm infants have dropped already by 50% after 48 h; in term infants, this postnatal decline occurs more slowly [10,24]. Plasma choline concentrations of preterm infants then remain at 2–3 times below the plasma choline concentration of a reference fetus. Animal models indicate that to maintain choline, phosphatidylcholine and lipoprotein metabolism of the liver, and possibly to enable brain growth [25], choline shortage induces hepatic choline accretion at the expense of other organs like the lungs [26,27].

The current standard diet of premature infants consists of breast milk enriched with multi-component supplements to meet the infants’ increased nutritional needs. However, an optimization of the choline concentration of both components may be necessary to address choline deficiency in VLBWI.

Choline content (median (p25–p75): 158 mg/L (61–360 mg/L)) of breast milk in mothers of preterm infants is lower than in mothers of term-born infants (258 mg/L (142–343 mg/L)), contrasting the higher metabolic needs of VLBWI. Whereas median milk content further decreases during lactation, short term concentrations are relatively stable and can be low or high [28]. The choline status of the mother, its content in breast milk and the availability of choline for enterally fed preterm infants are all influenced by oral choline intake of the mother and the genotypic variability of hepatic choline metabolism (e.g., frequent SNPs of the PEMT pathway, see above) [23,29]. We hypothesized that the differences in enteral choline intake resulting from variable intra-individual choline concentrations of breast milk, together with the low choline content of breast milk fortifier, may aggravate the choline deficit of preterm infants. To verify this hypothesis, we examined whether differences in total enteral choline intake (consisting of breast milk, breast milk fortifier and formula) are associated with choline, betaine and phosphatidylcholine plasma levels of VLBWI.

## 2. Materials and Methods

### 2.1. Study Population

This is a secondary analysis of data derived from a randomized controlled trial (RCT) [30] investigating the effect of different levels of protein intake on short term growth of VLBWI. The trial was registered at www.clinicaltrials.gov (NCT01773902 accessed on 18 January 2013), and the protocol approved by the Institutional Review Board. Written informed parental consent for human milk analysis, fortification and blood sampling was obtained. The study was conducted in accordance with the Declaration of Helsinki. To investigate the primary objective, 60 predominantly breast-milk-fed preterm infants (born at <32 weeks’ gestation, birthweight <1500 g) were included and randomized using a three-arm study design. Patients received either standardized low protein (fixed dose of standard fortifier in breast milk), standardized high protein (fixed dose of study fortifier) or supplemented breast milk based on actually measured breast milk protein content. No difference in growth velocity from birth to the end of intervention was shown (primary outcome). For this secondary analysis only 34 patients could be included due to a delayed start of breast milk collection in the underlying RCT.

### 2.2. Study Design (Secondary Analysis)

During the period of intervention, breast milk samples were collected twice weekly to measure choline-containing compounds. All infants received at least 100 mL/kg/d of enteral nutrition at the time of inclusion into the underlying RCT (median, day 7), and the first blood samples evaluated in this secondary analysis were taken on day 17 in median. Venous blood samples to measure plasma choline, betaine and PC concentrations were scheduled on days 14 (±2) and 28 (±4) after randomization. To avoid study-driven blood sampling, the latter was re-scheduled until a clinical indication arose. Choline content of the closest breast milk sample (collected at median 1 day (p25–p75: −1–+1 day) before blood sampling) was used to calculate actual enteral choline intake (comprising breast milk choline content; choline content of the administered standard—or study fortifier; protein supplement and formula) on the day before blood sampling. Only one breast milk–plasma pair per patient was included, which was always the first available breast milk–plasma pair after birth, for which the time difference between breast milk and blood sample was less than 5 days.

### 2.3. Chemical Analysis

Chemicals:

Chloroform (HPLC grade) was from Baker (Deventer, The Netherlands). Methanol, acetonitrile and water (analytical grade) were from Fluka analytical/Sigma Aldrich (Munich, Germany). Standards (choline chloride (>99%) and betaine monohydrate (>99%)) were from Sigma-Aldrich (Munich, Germany). D4-choline chloride (choline-1,1,2,2-d4) as an internal standard was purchased from C/D/N Isotopes Inc. (Pointe-Claire, QC, Canada) and 1,2-diarachidoyl-sn-gylcero-3-phosphocholine as an internal standard was purchased from Avanti Polar Lipids, Inc. (Alabaster, AL, USA). The purity of the chemicals was checked by tandem mass spectrometry (see below). All other chemicals were of analytical grade and from various commercial sources.

Extraction:

Plasma and milk samples were extracted according to Bligh and Dyer [31]. Glassware for sample extraction was cleaned with methanol to avoid contamination of samples with plasticizers or detergent.

#### 2.3.1. Breast Milk Samples

##### Sample Collection

An aliquot of 5 mL breast milk was collected twice weekly (Tuesdays and Fridays) after enrollment. Breast milk was heated to 40 °C and gently shaken to homogenize in preparation for feeding the premature infants, then aliquots were taken and stored at −30 °C until further measurement. The time of day at which the milk was obtained from the mother was not taken into account, since the choline content of breast milk (water and fat soluble) does not vary systematically in this respect [32].

##### Sample Extraction

A volume of 100 µL D4-choline (150 µmol/L in water) was added to a 100 µL sample volume as an internal standard, as well as 5 µL dibutylhydroxytoluene (BHT) solution (20 mg/mL in ethanol) as an antioxidant. Afterwards, 0.6 mL water, 2.4 mL methanol and 0.8 mL chloroform were added and the mixture stirred to achieve a ternary mixture kept at 4 °C for 1 h. To separate hydrophilic from lipophilic compounds, 1.6 mL water and 2.4 mL chloroform were added, and the sample was vigorously stirred and centrifuged at 3000× *g* at 4 °C for 20 min. This resulted in an upper phase (4.8 mL) containing choline, water-soluble choline esters (glycerophosphocholine, phosphocholine) and betaine, and a lower phase (about 3.2 mL), adjusted to 4 mL, containing the phospholipids (phosphatidylcholine, lyso-phosphatidylcholine, sphingomyelin). Samples were stored at −80 °C.

#### 2.3.2. Plasma Samples

##### Sample Collection

Venous blood samplings were scheduled on days 14 (±2) and 28 (±4) according to the study protocol of the underlying RCT, but had to be rescheduled until a clinical indication for blood sampling arose to avoid study-driven venous punctures for ethical reasons. Time deviation from milk samples were −5 to +1 days. EDTA blood was centrifuged to separate EDTA plasma within 30 min of venipuncture, with 50 µL plasma immediately frozen at −30 °C and stored at −80 °C until analysis.

##### Sample Extraction

A volume of 25 µL D4-choline (30 µmol/L in water), 0.35 mL water, 1.2 mL methanol and 0.4 mL chloroform were added to 25 µL ample volume and stirred. Afterwards 2.5 µL BHT solution (2.5 mg/mL in ethanol) and 10 µL 1,2-diarachidoyl-sn-glycero-3-phosphocholine (500 µmol/L in trifluoroethanol) was added and the ternary mixture was kept at 4 °C for 1 h. In order to separate the phases, 0.8 mL water and 1.2 mL chloroform were added, then the mixture was vigorously stirred and centrifuged to achieve an upper phase with choline, glycerophosphocholine, phosphocholine and betaine, and a lower phospholipid phase (see above). The upper phase (about 1.6 mL) was adjusted to 2 mL and stored at −80 °C.

#### 2.3.3. Analysis of Breast Milk and Plasma Samples

Electrospray ionization tandem mass spectrometry (ESI MS/MS) was used to analyze choline, glycerophosphocholine, phosphocholine and betaine as described before (2015) [10]. Equipment used included a Finnigan Surveyor Autosampler Plus, a Finnigan Surveyor MS Pump Plus and a TSQ Quantum Discovery MAX equipped with a heated electrospray ionization interface (H-ESI) (Thermo Fisher Scientific, Dreieich, Germany). Equally, phospholipids were determined by electrospray ionization tandem mass spectrometry (ESI-MS/MS) as previously described [33].

### 2.4. Statistical Analysis

Statistical analyses were performed using Microsoft Excel and JMP (SAS Institute, Cary, NC, USA, Version 16). Data are shown as median and interquartile range (p25–p75). For assessment of correlations between enteral intake and plasma concentrations, Spearman’s correlation coefficient *ρ* is reported. To enable an estimation of the plasma levels that could be expected with various enteral choline intakes, regression lines of linear regression analyses are depicted in the scatter plots.

### 2.5. Choline Content of the Multicomponent Fortifiers and Preterm Formula and Calculation of Total Enteral Choline Intake

Table 1 shows choline content of the fortifiers, the protein supplement and the administered preterm formula measured by ESI-MS/MS as described above and previously reported [34].

Total enteral choline intake was calculated as mg/kg of choline equivalent for the day before blood sampling based on the actual intake of breast milk, fortifier and formula, where breast milk total choline content was taken from the nearest available breast milk sample, since the choline content of breast milk is relatively constant for these short periods of time [28].

## 3. Results

### 3.1. Demographics

Demographic data of the included preterm infants are shown in Table 2. In total, there were 26 singleton pregnancies and 7 multiple pregnancies, but only one mother gave permission for both twins to be included in the study and randomized separately. In all other twins, only the first born was included because parents wished that all siblings were treated equally.

### 3.2. Choline Supply

An overview of all breast milk data concerning concentrations of choline and its metabolites in the underlying study was published [28].

The specific breast milk samples investigated in this secondary analysis that temporally matched the blood samples showed a median (p25–p75) total choline content of 171.7 (142.9–230.6) mg/L consisting of water-soluble choline components (choline, phosphocholine and glycerophosphocholine), phosphatidylcholine (PC), lyso-phosphatidylcholine (Lyso-PC) and sphingomyelin (SPH) (Table 3). One patient had only received formula on the day before the blood sample was taken.

Study patients had received a median (p25–p75) of 41.1 (33.8–45.9) mg/kg choline equivalent (at a median enteral intake of 165.2 (156.1–171.4) mL/kg/d of fortified breast milk) the day before blood sampling was performed. The administered choline was derived to 69% (62–77) from breast milk and to 24% (20–32) from breast milk fortifier (proportion of formula/protein supplement was negligible).

### 3.3. Plasma Concentrations of Choline, Betaine and Choline Compounds

Blood samples were taken at 16 (14–19) days after birth and a postmenstrual age of 32.7 (32.0–33.6) weeks. The medians (p25–p75) of the plasma concentrations of choline, betaine and the choline compounds are shown in Table 4.

### 3.4. Correlation of Choline Intake with Water-Soluble Choline Components and Betaine Plasma Concentrations

Plasma concentrations of betaine and choline showed a significant positive correlation (*ρ* = 0.48; *p* = 0.0037) (Figure 1).

There was no significant correlation between total enteral choline intake (mg/kg/d) and plasma concentration (µmol/L) of the sum of water-soluble choline components (choline, phosphocholine and glycerophosphocholine) (*ρ* = 0.21; *p* = 0.25) (Figure 2a), but there was a significant positive correlation between total choline intake and plasma betaine concentrations (*ρ* = 0.55; *p* = 0.0007) (Figure 2b). Similarly, a significant correlation was demonstrated when comparing total choline intake with the sum of choline and betaine plasma concentrations (*ρ* = 0.55; *p* = 0.0007) (Figure 2c).

### 3.5. Correlation of Choline Intake with Phosphatidylcholine and Sphingomyelin Concentrations

In contrast to the water-soluble compounds, no correlations were found between total enteral choline intake and the plasma concentrations of phosphatidylcholine (*ρ* = 0.06; *p* = 0.75)*,* lyso-phosphatidylcholine (*ρ* = 0.01; *p* = 0.96) and sphingomyelin (*ρ* = 0.02; *p* = 0.91) (Figure 3a–c).

## 4. Discussion

Choline and its metabolites play a central role in parenchymal homeostasis and growth, osmoregulation, methyl group transfer, liver and lipoprotein metabolism and, notably, neurologic development [2,3,4,7,8,9]. Therefore, sufficient choline supply seems indispensable to achieve adequate growth and development.

Presumably due to the high growth rate of the fetus between 24- and 34-weeks postmenstrual age and the resulting higher nutrient requirements, exceptionally high choline plasma levels were observed in cord blood (45 (36–60) µmol/L), as a surrogate for fetal blood, during this period. These levels are higher than after 34 weeks (37 (29–45) µmol/L) and, particularly, than in adults (10 (8–12) µmol/L) [10,35].

Following premature birth, plasma levels fall to half the cord blood concentrations within 48 h and remain at this low concentration thereafter or decrease even further [10]. Because cellular uptake of choline is directly proportional to its plasma concentrations (~8–80 µmol/L), lower plasma concentrations result in a decrease in intracellular choline availability for phosphatidylcholine synthesis via phosphocholine as an intermediate [36]. Moreover, availability of betaine, as an oxidation product of choline used to form methionine from homocysteine, is decreased as well [37]. In its activated form, i.e., S-adenosylmethionine, it is an essential methyl group donor throughout the organism [38], including the pathway for endogenous choline synthesis, i.e., synthesis of phosphatidylcholine via threefold methylation of phosphatidylethanolamine (remodeling pathway) by phosphatidylethanolamine-N-methyl-transferase (PEMT).

Preclinical studies in rats showed improved neurocognitive development in the offspring after maternal choline supplementation, suggesting that optimized maternal choline supply may be important for infants as well [39,40]. Fast growing newborn rats have choline plasma concentrations similar to the human fetus, and choline supplementation results in both increased plasma choline and betaine [26].

This secondary analysis of an RCT shows that this not only occurs in response to choline supplementation, but also under basic conditions of nutritional choline supply by feeding fortified breast milk to human preterm infants, suggesting that variable choline supply with breast milk impacts both choline as well as betaine/methyl group status of VLBWI [10]. These findings agree with our recent studies of enteral choline supplementation in premature infants [36,41] and indicate that both choline and betaine plasma concentrations, including the methyl group status, can be improved by enteral choline supplementation. Consequently, enteral choline supplementation offers the possibility to compensate for the cumulative choline deficit of the premature infant caused by premature discontinuation of the continuous active supply via the placenta [13]. The fact that enteral total choline supply better correlates with betaine than choline concentrations may be attributable to the slower turnover of betaine, resulting in a more sustained increase in plasma concentrations following enteral supplementation [41]. In addition, the sustained increase in betaine levels suggests that an important aspect of any choline supplementation is feeding the one-carbon pool via betaine, which may help to increase endogenous choline synthesis e.g., via phosphatidylethanolamine N-methyltransferase (PEMT). In contrast to the correlations between enteral choline intake as well as choline and betaine plasma concentrations, this study showed no significant correlation between enteral choline intake and plasma concentrations of phosphatidylcholine. However, plasma turnover of choline, betaine and phospholipids are highly different, with choline showing the fastest plasma increase and turnover upon administration (t1/2 <1 h), followed by betaine and phospholipids [42]. As plasma phospholipids predominantly represent lipoprotein metabolism (for review, see [43]), differences may depend on hepatic and other organs’ function rather than directly on short-term choline supply.

The remaining question is how much choline premature infants should be supplemented with to achieve normal growth and development. From our previous short-term supplementation study it appears that an extra enteral intake (supplementation in addition to milk and standard fortifier) of around 30 mg/kg/d of total choline equivalent is required to achieve near-fetal plasma levels of 35–45 µmol/L [36]. The regression analysis performed in this study (Figure 2a, comparing enteral choline intake in mg/kg/d and choline plasma concentrations) is consistent with our earlier findings and assumptions, indicating that an enteral choline intake >60 mg/kg/d is necessary to achieve fetal-like choline plasma concentrations >35 µmol/L.

Patients in the present analysis had a median enteral choline intake of 40 mg/kg/d, and thus an intake at the upper end recommended by the European Society for Pediatric Gastroenterology, Hepatology and Nutrition (ESPGHAN) (8–55 mg/kg/d) [44]. Nevertheless, median plasma levels of 20 µmol/L at a gestational age of 32 weeks were reduced by 50% compared to those of a reference fetus, where ~32 weeks gestation corresponds to plasma choline concentrations of 41.4 (31.8–51.2) µmol/L [10], signaling insufficient supply. Consequently, current recommendations for choline intake of preterm infants with <34 weeks’ gestation appear inadequate, and their actual needs probably are at the upper limit of ESPGHAN recommendation or even higher. In an effort to determine the optimal choline supply, we suggest to take the growth spurt of preterm infants into account, by comparing their growth rate to that of term infants and to extrapolate their AI accordingly (term infants: 18 mg/kg/d) [12], as is usually determined for protein requirements. An extrapolation describing adequate intake as a linear function of growth results in a suggested supply of 70–85 mg/kg/d [12,13,14,15], which is consistent with the finding that an extra 30 mg/kg/d of choline on top of fortified breast milk resulted in near-fetal plasma levels (35.4 (32.8–41.7) µmol/L) in VLBWI [36].

Considering that, in spite of near-in-utero weight gain, VLBWI still show deficits in fat free mass and brain growth compared to term infants [45,46], it seems most likely that current nutritional guidelines do not yet meet infants’ actual needs. The positive effect of protein supplementation on growth shows a ceiling effect at ~4.5 mg/kg/d [30]. In addition, preterm infants have an increased fat mass and decreased lean body mass at term-equivalent age [47], indicating enough or even excessive energy supply. It therefore seems likely that deficiency in an essential nutrient, as a structural and functional component of tissues—mainly in the form of membrane phosphatidylcholine—may be one limiting factor, beyond optimized protein supply and energy intake for the growth of premature infants. However, we were unable to demonstrate an association between choline intake and weight gain or head circumference growth in this small cohort.

In this secondary analysis, about two thirds of enteral choline intake occurred through breast milk, and one third through breast milk fortifier. Choline concentration in breast milk shows relevant inter- and intra-individual variability, and a choline deficit could be underestimated if based on single measurements. Individual variability may arise from SNPs of enzymes involved in choline metabolism as well as variability in maternal choline intake (which is compromised in a substantial proportion of parturients) [23]. Given that low maternal plasma choline levels and breast milk concentration, as well as lower oral intake by the infant, were associated with impaired cognitive outcome [48], adequate nutrition and/or choline supplementation of parturients may be considered to improve enteral choline intake of breast fed infants. Hence, optimization of the choline status of pregnant women, to ensure optimal choline supply via the placenta and breast milk, might be necessary. Moreover, introducing a higher choline concentration of common breast milk fortifiers is also necessary, since the current supply via breast milk supplements does not result in physiological plasma concentrations of choline. These needs have to be addressed in subsequent investigations [32].

Limitations: This study is a secondary analysis, and therefore only allows describing associations rather than causal relationships. Due to the protocol of the original study, choline content of breast milk was only measured twice a week. Therefore, breast milk data were not always available exactly on the day prior to blood sampling. However, to make a comparison possible, the breast milk sample taken closest to blood sampling was matched with the blood sample, which resulted in a proportion of 85% measured at ±1 day before blood sampling (max. 5 days).

## 5. Conclusions

Choline intake showed significant positive correlations with plasma levels of combined choline and betaine as well as betaine in predominantly breast-fed very low birth weight infants. Current feeding strategies do not seem to meet the choline needs of preterm infants as resulting plasma concentrations are only half those of corresponding fetal levels. Adequate intake of choline in VLBWI might be higher than previously assumed, and may even exceed the upper limit of ESPGHAN recommendations (55 mg/kg/d). However, whether or not enteral choline supplementation aiming for fetal-like choline plasma concentrations does indeed improve growth and outcome of preterm infants yet remains to be studied.

## Figures and Tables

**Figure 1 nutrients-15-04758-f001:**
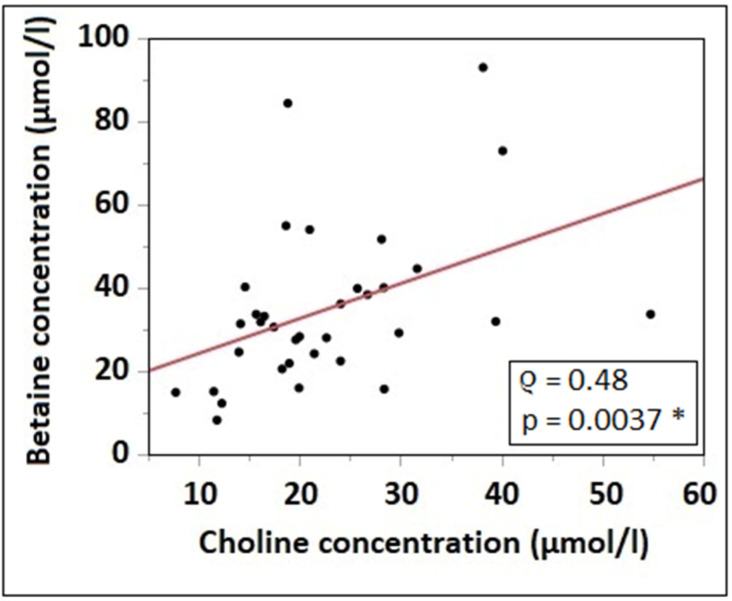
Correlation of plasma choline (µmol/L) with plasma betaine concentration (µmol/L); regression line from linear regression analysis. * indicates a significant correlation.

**Figure 2 nutrients-15-04758-f002:**
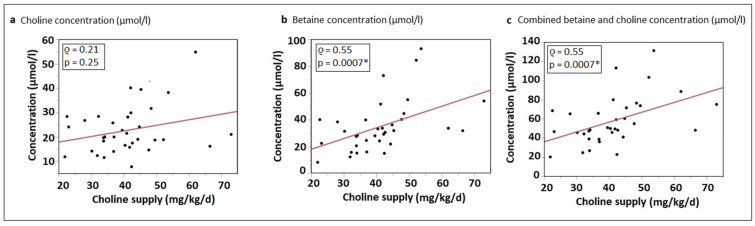
Correlation of enteral intake of total choline equivalent (mg/kg/d) and (**a**) plasma water-soluble choline concentration (µmol/L); (**b**) plasma betaine concentration (µmol/L); and (**c**) combined plasma choline and betaine concentration (µmol/L); regression lines from linear regression analyses. * indicates a significant correlation.

**Figure 3 nutrients-15-04758-f003:**
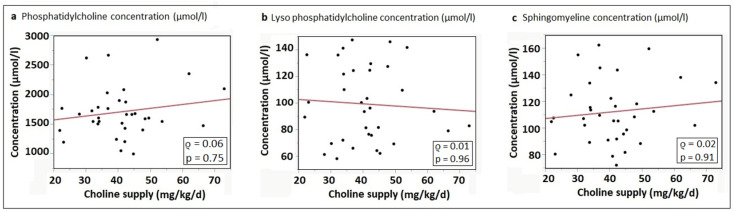
Correlation of enteral choline intake (mg/kg/d) and (**a**) plasma phosphatidylcholine concentration (µmol/L); (**b**) plasma lyso-phosphatidylcholine concentration (µmol/L); and (**c**) plasma sphingomyelin concentration (µmol/L); regression lines from linear regression analyses.

**Table 1 nutrients-15-04758-t001:** Choline content of used fortifiers and formula shown as mg/g or mL.

	Total Choline Equivalent Content *	Administered Choline at Standard Full Feeds (150 mL/kg/d)
FM 85 (before 2017) (standard fortifier)	1.40 mg/g	10.5 mg/kg/d
10.01.DE.INF (study fortifier)	1.54 mg/g	11.55 mg/kg/d
BEBA FN1 (preterm formula <1800 g)	0.125 mg/mL	18.75 mg/kg/d
Aptamil Eiweiß + (protein supplement)	0.09 mg/g	-

* comprising all water soluble and lipid soluble choline containing compounds converted to choline equivalent.

**Table 2 nutrients-15-04758-t002:** Demographic data of the infants included in this secondary analysis.

Number of mothers	33
Number of singleton pregnancies	26 (79%)
Number of infants	34
PMA at birth (weeks)	30.21 (29.11–31.14)
Sex (m/f)	19/15
Birth weight (kg)	1.18 (1.04–1.39)
Postnatal age at time of blood sampling (days)	16 (14–19)
PMA at time of blood sampling (weeks)	32.71 (32.04–33.57)
Number of infants n (%) with breast milk sample taken relative to blood sample	
−5 days	1 (3%)
−4 days	0 (0%)
−3 days	3 (9%)
−2 days	1 (3%)
−1 day	9 (27%)
0 day	11 (32%)
+1 day	9 (27%)

Values are median (p25–p75) or number (%); sum of % values may exceed 100% due to rounding.

**Table 3 nutrients-15-04758-t003:** Breast milk content of choline equivalent (mg/L) from different choline compounds in human breast milk.

Compound	Choline Equivalent * (mg/L) Median (p25–p75)	Choline Administered at Full Feeds (150 mL/kg/d)	Fraction (%)
Water-soluble choline components:	146.1 (115.1–194.3)	21.9 mg/kg/d	84.5
Thereof:			
- Free choline	17.2 (12.8–24.4)	2.6 mg/kg/d	
- Glycerophosphocholine	65.4 (44.5–84.8)	9.8 mg/kg/d	
- Phosphocholine	67.1 (52.4–96.4)	10.1 mg/kg/d	
Betaine	0.6 (0.3–1.0)	0.1 mg/kg/d	-
Lipid-soluble choline components			
- Phosphatidylcholine	20.1 (14.3–22.8)	3.0 mg/kg/d	11.6
- Lyso-phosphatidylcholine	2.0 (1.0–3.0)	0.3 mg/kg/d	1.2
- Sphingomyelin	4.8 (3.9–5.8)	0.7 mg/kg/d	2.8

Values are median (p25–p75) and % of total choline content. * to convert from choline equivalent in mg/L to µmol/L, multiply with 9.62.

**Table 4 nutrients-15-04758-t004:** Plasma concentrations of the different choline compounds and metabolites.

Compound	Plasma Concentrations (µmol/L) Median (p25–p75)	Fraction (%)
Water-soluble choline components:	20.0 (16.0–28.2) µmol/L	1.1
Thereof:		
- Free Choline	18.2 (15.4–25.9) µmol/L	
- Glycerophosphocholine	0.31 (0.12–0.73) µmol/L	
- Phosphocholine	0.40 (0.17–0.85) µmol/L	
Betaine	31.6 (22.3–40.1) µmol/L	-
Lipid-soluble choline components		
- Phosphatidylcholine	1630.7 (1460.2–1876.5) µmol/L	88.0
- Lyso-phosphatidylcholine	94.9 (74.7–125.2) µmol/L	5.1
- Sphingomyelin	107.8 (94.4–126.9) µmol/L	5.8

Values are median (p25–p75) and % of total choline content in plasma; sum of % values may exceed 100% due to rounding.

## Data Availability

Data are contained within the article.

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
