# Peer review of "Choline and Betaine Levels in Plasma Mirror Choline Intake in Very Preterm Infants"

_nutrients, 2023, doi:10.3390/nu15224758_

Round 1

Reviewer 1 Report

Comments and Suggestions for Authors

The manuscript by Minarski et al „Choline and Betaine levels in plasma mirror choline intake in very preterm infants“ provides valuable information and is good to read. Besides the secondary presentation of data it includes a very interesting introduction and discussion.

There are only a few points I would like to mention

Line 138ff: from the description of milk sampling it is not clear if samples were taken in a way to be representative for the fat content, which seems important as PC and SM are major lipids in milk fat globule membranes

Line 163: sample

Line 178ff: as medians are given, it seems that data were not normally distributed, please indicate how it was tested that data are suitable for linear regression and Pearson correlation

Line 196ff: if only one twin was included, how was the decision made?

Table 2: it seems the molar ratio between PC and SM was around 4:1, which seems not typical for mature milk for term infants, where PC and SM are more similar. This should be discussed

Line 289: please provide a reference

Line 318: this statement is not clear from my point of view, please provide a more detailed explanation

Line 333: “resulted in near-fetal plasma levels” is not clear to me

Reviewer 2 Report

Comments and Suggestions for Authors

This is a secondary analysis of a study on the supplementation of feeding with  choline for preterm infants. Unfortunately, I could not open ref 35 and 38 where the original study is described, so I can not evaluate if the results of this secondary analysis are influenced by the randomization for the original study. 

This research group has published a number of papers regarding the topic of choline needs of newborn infants. They claim in the different papers that the choline intake of preterm infants is too low. The hypothesis of the present paper is if differences in enteral choline intake resulting from variable intra-individual choline concentrations in breastmilk, together with the low choline content of breast milk fortifier, may aggravate the choline deficit of preterm infants. 

The conclusion of the study is that the combined level of plasma choline and betaine are correlated with choline intake, but the intake of choline is not correlated with the plasma level of choline. The intake was insufficient to achieve fetus like levels of choline. Higher intake might be needed to meet the choline requirements of preterm infants.

The question and results raise some important questions.

1. How to define the desired choline level for  a preterm infant? Is the level in the umbilical cord the norm, how is this influenced by the choline concentration in the mother? Or, is the betaine level indicative, might a high betaine level indicate a surplus in choline, thereby inducing the choline methionine pathway. In other words, is betaine needed or a degradation product? This relates to the lines 302 and following, can this be an explanation why betaine is related to the intake, while choline is not? As long as the optimal level of choline is not defined, is it difficult to discuss the optimal intake. This issue needs more discussion.

2 it is unclear how long the infants received enteral feeding before measuring the first choline level. A difference in time might have influenced the results. A delay in enteral feeding is often seen in the most sick infants.

3. Was there a correlation between choline intake and increase in length and head circumference?  The authors write that choline is correlated with lean body mass. No difference in weight gain does not mean no difference in lean body mass and HC. A difference in lean body mass is much more important than a difference in weight gain.

4. what is the explanation for the finding that betaine, but not choline, is related to the choline intake?

5. I  am not sure I believe that the results of this study indicate that the choline intake must be higher than presently advised. 

6. Is there a correlation between cord plasma levels of choline and gestational age? That might help defining the optimal plasma choline level in a preterm infant. 
